# Hyperglycemia induced cathepsin L maturation linked to diabetic comorbidities and COVID-19 mortality

Qiong He[1†], Miao-Miao Zhao[1*†], Ming-Jia Li[1], Xiao-Ya Li[1], Jian-Min Jin[2], Ying-Mei Feng[3], Li Zhang[4], Wei Jin Huang[4], Fangyuan Yang[1], Jin-Kui Yang[1,5*]

[1]Department of Endocrinology, Beijing Diabetes Institute, Beijing Tongren Hospital, Capital Medical University, Beijing, China; [2]Department of Respiratory and Critical Care Medicine, Beijing Tongren Hospital, Capital Medical University, Beijing, China; [3]Department of Science and Technology, Beijing Youan Hospital, Capital Medical University, Beijing, China; [4]Division of HIV/AIDS and Sex-Transmitted Virus Vaccines, Institute for Biological Product Control, National Institutes for Food and Drug Control (NIFDC), Beijing, China, Beijing, China; [5]Laboratory for Clinical Medicine, Capital Medical University, Beijing, China

*For correspondence:
mmzhao@ccmu.edu.cn (M-MZ);
jkyang@ccmu.edu.cn (J-KY)

[†]These authors contributed equally to this work

**Abstract** Diabetes, a prevalent chronic condition, significantly increases the risk of mortality from COVID-19, yet the underlying mechanisms remain elusive. Emerging evidence implicates Cathepsin L (CTSL) in diabetic complications, including nephropathy and retinopathy. Our previous research identified CTSL as a pivotal protease promoting SARS-CoV-2 infection. Here, we demonstrate elevated blood CTSL levels in individuals with diabetes, facilitating SARS-CoV-2 infection. Chronic hyperglycemia correlates positively with CTSL concentration and activity in diabetic patients, while acute hyperglycemia augments CTSL activity in healthy individuals. In vitro studies reveal high glucose, but not insulin, promotes SARS-CoV-2 infection in wild-type cells, with *CTSL* knockout cells displaying reduced susceptibility. Utilizing lung tissue samples from diabetic and non-diabetic patients, alongside *Lepr*db/db mice and *Lepr*db/+ mice, we illustrate increased CTSL activity in both humans and mice under diabetic conditions. Mechanistically, high glucose levels promote CTSL maturation and translocation from the endoplasmic reticulum (ER) to the lysosome via the ER-Golgi-lysosome axis. Our findings underscore the pivotal role of hyperglycemia-induced CTSL maturation in diabetic comorbidities and complications.

## eLife assessment

This **valuable** study advances our understanding of why diabetes is a risk factor for more severe Covid-19 disease. The authors offer **convincing** evidence that cathepsin L is more active in diabetic individuals because of the presence of high glucose, where the main mechanism is increased cathepsin L maturation. This study should be of interest to researchers in diabetes, virology and immunology.

## Introduction

Cysteine proteases, including cathepsin L (CTSL), hold pivotal roles in human pathobiology due to their multifaceted activities within and outside cells. Emerging evidence links cathepsins, notably CTSL, to metabolic disorders like obesity and diabetes, as well as diabetic complications (*Crawford et al., 2022*; *Ding et al., 2020*; *Limonte et al., 2022*). Previous studies have associated CTSL with

**eLife digest** People with diabetes are at greater risk of developing severe COVID-19 and dying from the illness, which is caused by a virus known as SARS-CoV-2. The high blood sugar levels associated with diabetes appear to be a contributing factor to this heightened risk. However, diabetes is a complex condition encompassing a range of metabolic disorders, and it is therefore likely that other factors may contribute.

Previous research identified a link between an enzyme called cathepsin L and more severe COVID-19 in people with diabetes. Elevated cathepsin L levels are known to contribute to diabetes complications, such as kidney damage and vision loss. It has also been shown that cathepsin L helps SARS-CoV-2 to enter and infect cells. This raised the question of whether elevated cathepsin L is responsible for the increased COVID-19 vulnerability in patients with diabetes.

To investigate, He, Zhao et al. monitored disease severity and cathepsin L levels in patients with COVID-19. This confirmed that people with diabetes had more severe COVID-19 and that higher levels of cathepsin L are linked to more severe disease. Analysis also revealed that cathepsin L activity increases as blood glucose levels increase.

In laboratory experiments, cells exposed to glucose or fluid from the blood of people with diabetes were more easily infected with SARS-CoV-2, with cells genetically modified to lack cathepsin L being more resistant to infection. Further experiments revealed this was due to glucose promoting maturation and migration of cathepsin L in the cells.

The findings of He, Zhao et al. help to explain why people with diabetes are more likely to develop severe or fatal COVID-19. Therefore, controlling blood glucose levels in people with diabetes may help to prevent or reduce the severity of the disease. Additionally, therapies targeting cathepsin L could also potentially help to treat COVID-19, especially in patients with diabetes, although more research is needed to develop and test these treatments.

proteinuria in podocytes (*Reiser et al., 2004*) and intraocular angiogenesis (*Shimada et al., 2010*), suggesting its potential as a therapeutic target for diabetic nephropathy and vision-threatening conditions such as proliferative diabetic retinopathy (*Shimada et al., 2010*).

Recent investigations highlight CTSL's involvement in the cleavage and processing of the SARS-CoV-2 spike protein, critical for viral entry and replication within host cells, as reported by our group (*Zhao et al., 2021b*) and others (*Jackson et al., 2022*; *Muralidar et al., 2021*). Our previous studies show that elevated CTSL levels correlate with disease severity and CTSL is crucial for activation of all emerging SARS-CoV-2 variants, making it a potential drug target for future mutation-resistant therapy (*Zhao et al., 2021b*; *Zhao et al., 2022*). However, the specific role of CTSL in COVID-19 infection among diabetic patients remains unexplored.

Patients with diabetes face heightened risks of severe COVID-19 outcomes and increased mortality rates (*Khunti et al., 2021*). Studies report a 1.23–5.87 times higher likelihood of severe COVID-19 and death among diabetic individuals compared to non-diabetic counterparts (*Dennis et al., 2021*; *Shi et al., 2020*; *Williamson et al., 2020*). Diabetes ranks as the second most prevalent chronic comorbidity contributing to COVID-19 fatalities, following hypertension, according to data from the American Centers for Disease Control and Prevention (CDC). Notably, within the diabetic cohort, individuals with elevated blood glucose levels (HbA1c≥7.5%) exhibit a higher hazard ratio for adverse outcomes compared to those with lower glucose levels (HbA1c<7.5%; *Williamson et al., 2020*).

This study delves into the impact of high glucose levels on CTSL maturation and its implications for diabetic comorbidities, complications, and susceptibility to SARS-CoV-2 infection. Our findings unveil the role of high glucose in promoting CTSL maturation and translocation from the endoplasmic reticulum to the lysosome, potentially exacerbating diabetic complications and contributing to COVID-19 susceptibility among diabetic individuals.

## Results

### Diabetic COVID-19 patients have severe conditions and elevated CTSL

In COVID-19 patients, we conducted a case-control study to examine the association of diabetes and COVID-19 severity in 207 COVID-19 inpatients from two hospitals. We matched 62 patients by gender and age, 31 with diabetes and 31 without (*Figure 1a*). *Supplementary file 1* summarizes the demographic and clinical characteristics of these diabetic and non-diabetic COVID-19 patients. We found that diabetic patients had a significantly higher risk of developing severe COVID-19 than non-diabetic patients according to the clinical classification criteria (http://www.nhc.gov.cn/), and had more symptoms such as fever, cough, fatigue, and dyspnea (*Figure 1b*). Diabetic COVID-19 patients showed higher levels of inflammation and infection markers (*Supplementary file 1*). These results suggest that diabetes is strongly associated with severity of COVID-19.

To explore the mechanisms of hyperglycemia and SARS-CoV-2 infection, we collected plasma samples from Beijing Youan Hospital on Day 0 (the admission day), Day 14, and Day 28 after discharge from the hospital (*Figure 1c*). SARS-CoV-2 infects host cells through the virus spike protein binding with ACE2 receptor. It uses host proteases, such as CTSL and cathepsin B (CTSB) to activate its spike protein by cleavage, which enhances its cell entry (*Jackson et al., 2022*; *Muralidar et al., 2021*). We measured the plasma levels of COVID-19 related proteins, ACE2, CTSL, and CTSB in diabetic and non-diabetic COVID-19 patients. Only CTSL levels were significantly higher in diabetic patients than in non-diabetic patients and changed with the course of COVID-19. CTSL peaked on admission day and decreased significantly after discharge from the hospital (*Figure 1d–f*). These results indicate that CTSL is strongly associated with COVID-19, as previously reported (*Zhao et al., 2022*), and may be involved in diabetes in COVID-19 patients.

### Impact of chronic and acute hyperglycemia on CTSL activity

In non-COVID-19 participants, we investigated the correlation of CTSL with chronic and acute hyperglycemia using two studies. First, to examine the impact of chronic hyperglycemia on CTSL, we performed a case-control study in 61 patients with type 2 diabetes and 61 euglycemic subjects, matched for sex and age (*Supplementary file 2*). We found that plasma CTSL activity was strongly positively correlated with chronic hyperglycemia indicated by HbA1c, and was significantly higher in diabetic patients than in euglycemic individuals (*Figure 2a and c*). Additionally, plasma CTSL concentration showed a positive trend with chronic hyperglycemia indicated by HbA1c (*Figure 2b and d*). Second, to examine the impact of acute hyperglycemia on CTSL, we performed a hyperglycemic clamp study in 15 healthy male subjects (*Figure 2e–g*). We observed that CTSL activity increased parallelly with blood glucose levels (*Figure 2h and i*). However, CTSL concentration did not change with blood glucose levels (*Figure 2j*). Therefore, chronic hyperglycemia is strongly associated with both CTSL concentration and activity, while acute hyperglycemia only affects CTSL activity.

We also observed a strongly correlation between CTSL activity/concentration and other chronic comorbidities like hypertension and coronary heart disease (CHD), especially diabetes (*Supplementary file 3*). Additionally, elevated blood glucose levels also accompanied with an increase in insulin and proinsulin C-peptide levels in acute hyperglycemic individuals (*Figure 2f and g*) and diabetic patients (insulin resistance; *Gerich, 2003*). It is unclear whether the increased CTSL activity/concentration is solely a result of hyperglycemia or the corresponding hyperinsulinemia, and further clarification is needed.

### Hyperglycemia enhances SARS-CoV-2 infection through CTSL

To validate our hypothesis that diabetic patients are more susceptible to SARS-CoV-2 infection results from higher CTSL levels, we conducted in vitro experiments using a SARS-CoV-2 pseudovirus system, which indicates the virus invasion rate instead of replication, and utilized the SARS-CoV-2 susceptible human hepatoma cell line Huh7, given that the liver is a primary target of SARS-CoV-2 (*Gupta et al., 2020*).

First, to examine if cells cultured with serum from diabetic patients are more susceptible to SARS-CoV-2 infection, we compared the infectivity of cells cultured with healthy and diabetic sera exposed to SARS-CoV-2 using a luciferase assay. The results showed that cells cultured with diabetic serum had higher infection rate than cultured with healthy serum (*Figure 3a*), indicating diabetic serum facilitated SARS-CoV-2 entry into host cells.

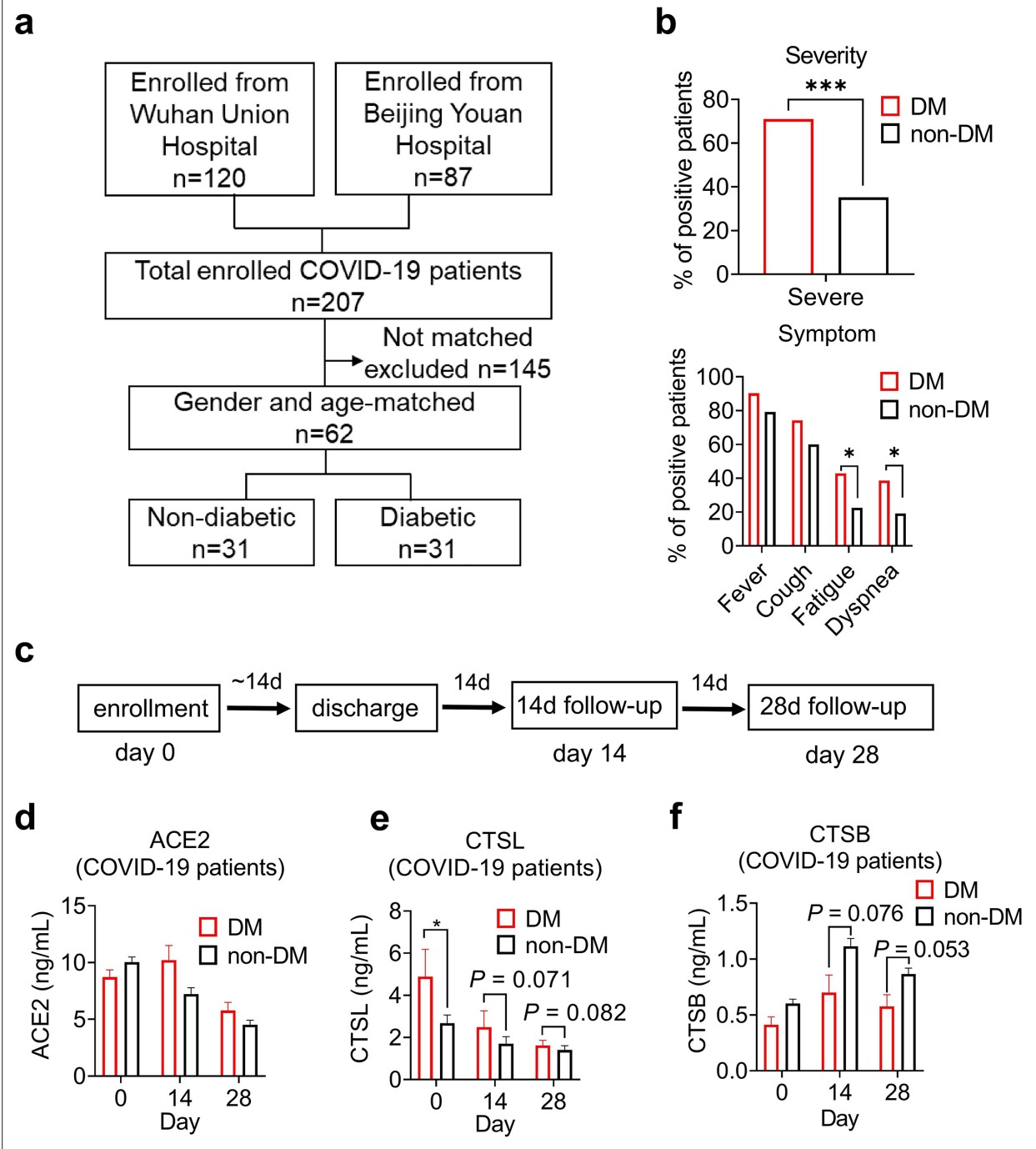

**Figure 1.** Disease severity and CTSL levels in COVID-19 patients with or without diabetes. (**a**) Design and inclusion flowchart of the case-control study. Out of 207 COVID-19 patients from two hospitals, 62 were included in the study after matching for gender and age, 31 with diabetes and 31 without. (**b**) Comparison of symptom severity and prevalence between diabetic and non-diabetic COVID-19 patients. (**c**) Study design and timeline of the enrollment and follow-up study. After admitted to the hospital (Day 0), patients were hospitalized for a mean duration of 14 days, followed up 14 days (Day 14) and 28 days (Day 28) after discharge, and blood samples were collected at each time point. (**d-f**) Plasma levels of COVID-19-related proteins were measured in diabetic and non-diabetic COVID-19 patients on Day 0, Day 14, and Day 28. Statistical significance was assessed by unpaired $t$-test (**b**) and Mann-Whitney $U$-test (**d-f**). The data are presented as the means ± SEM. *p<0.05, ***p<0.001.

The online version of this article includes the following source data for figure 1:

**Source data 1.** The dataset of COVID-19 patients with or without diabetes.

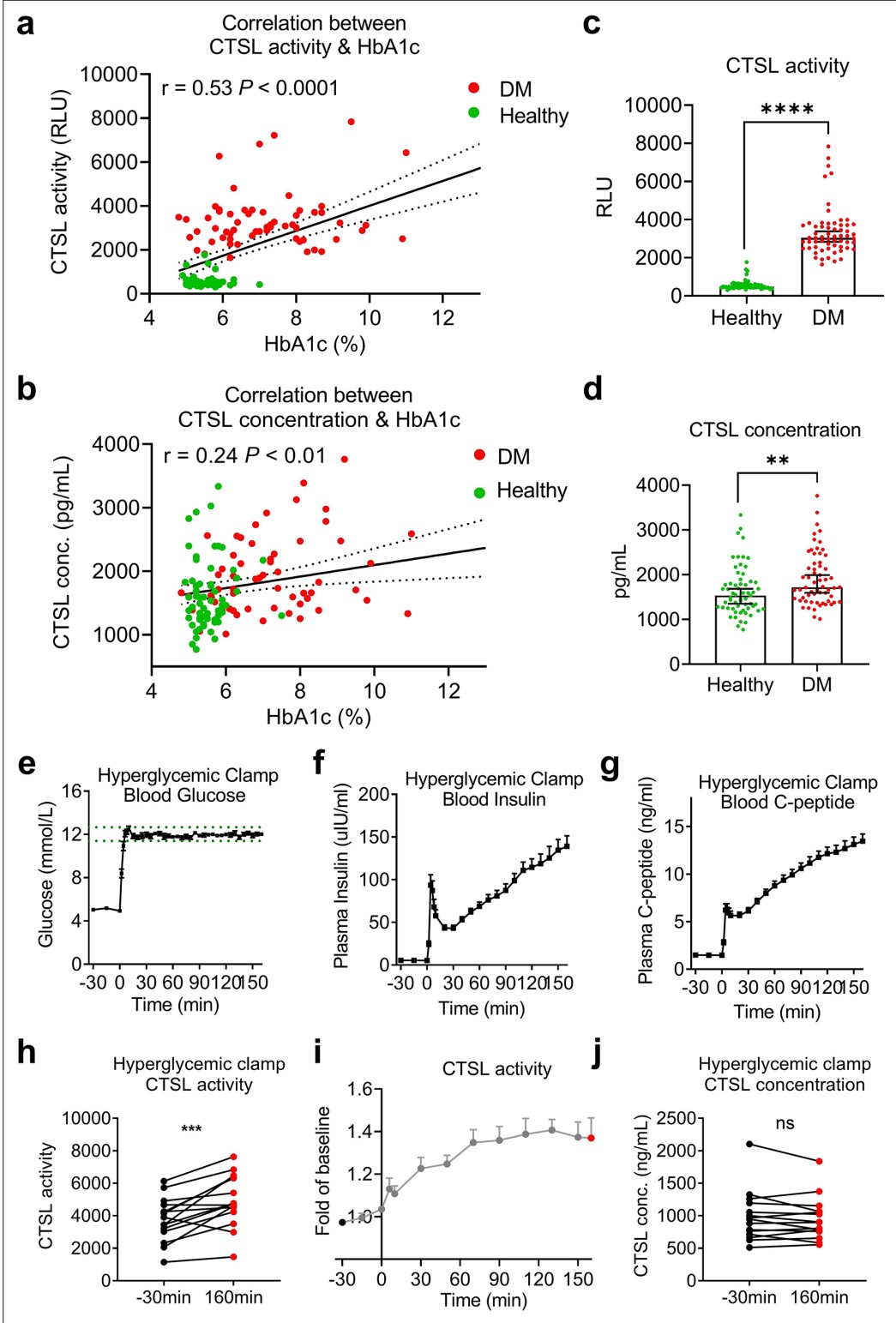

**Figure 2.** Impact of chronic and acute hyperglycemia on CTSL concentration and activity. (**a–d**) Effects of chronic hyperglycemia on CTSL activity and concentration in 122 gender- and age-matched individuals without COVID-19, including 61 euglycemic volunteers and 61 diabetic patients. (**a**) Correlation between plasma CTSL activity and blood glucose level indicated by HbA1c. (**b**) Correlation between plasma CTSL concentration and HbA1c. The dashed line represents the 95% CI in (**a and b**). (**c**) Comparison of plasma CTSL activity between the euglycemic and diabetic groups. (**d**) Comparison of plasma CTSL concentration between the euglycemic and diabetic groups.

*Figure 2 continued on next page*

*Figure 2 continued*

The data are presented as the median with 95% CI in (**c and d**). (**e-g**) Hyperglycemic clamp study performed in 15 healthy male subjects. (**e**) Plasma glucose levels in subjects throughout the clamp study. The dashed lines represent the range of ±5% of the hyperglycemic target level (basal blood glucose level +6.9 mmol/L). (**f**) Insulin levels and (**g**) proinsulin C-peptide levels throughout the clamp study. (**h–j**) Effects of acute hyperglycemia on CTSL concentration and activity in 15 healthy male volunteers. (**h**) Plasma CTSL activity at the beginning and the end of the clamp study. (**i**) Plasma CTSL activity throughout the clamp study. (**j**) Plasma CTSL concentration at the beginning and the end of the clamp study. Statistical significance was assessed by Spearman correlation analysis (**a and b**), unpaired *t*-test (**c and d**) and paired *t*-test (**h and j**). The data are presented as the means ± SEM. **p<0.01, ***p<0.001, ****p<0.0001.

The online version of this article includes the following source data for figure 2:

**Source data 1.** The dataset of healthy individuals with or without diabetes and the original data of the hyperglycemic clamp study.

Next, we investigated whether elevated blood glucose or insulin level promoted SARS-CoV-2 infection by culturing Huh7 cells in cell media with various concentrations of glucose or insulin. Consistent with our clinical data, the results showed that infection was more severe in Huh7 cells at high glucose levels, while insulin levels had a minimal impact on SARS-CoV-2 infection (*Figure 3b and c*).

Given that hyperglycemia increased CTSL levels (*Figure 2*) and facilitated SARS-CoV-2 entry into host cells (*Figure 3b*), we hypothesized that high blood glucose promoted SARS-CoV-2 infection through CTSL. To investigate the requirement of CTSL in SARS-CoV-2 infection, we used the CRISPR-Cas9 system to establish a stable *CTSL* knockout (KO) Huh7 cell line. The knockout efficiency of CTSL protein of *CTSL* KO Huh7 cell line was confirmed in *Figure 3d*. Compared with wild-type (WT) cells, knockout of *CTSL* led to a significant reduction in SARS-CoV-2 infection (*Figure 3e*), suggesting that CTSL is crucial for SARS-CoV-2 infection as we previously reported (*Zhao et al., 2022*).

We then conducted a *CTSL* KO Huh7 cell infection experiment under different glucose conditions, to illustrate the impact of glucose level on SARS-CoV-2 infection via CTSL. The results showed that WT Huh7 cell cultured in high-glucose medium exhibited a much higher infective rate than those in low-glucose medium. However, *CTSL* KO Huh7 cells maintained a low infective rate of SARS-CoV-2 regardless of glucose or insulin levels (*Figure 3f–h*). Therefore, hyperglycemia enhanced SARS-CoV-2 infection through CTSL. Considering that CTSL realizes its proteolytic function through its enzyme activity and protein concentration, subsequent studies aimed to reveal whether the CTSL activity or concentration changed under high-glucose condition.

## Elevation of glucose level boosts CTSL activity

Hyperglycemia can lead to metabolic acidosis and alter blood pH. However, the normal range for blood pH in humans is relatively narrow, typically ranging from 7.35 to 7.45. In our study, blood pH remained within this normal range for both diabetic and healthy control samples. *Figure 4a* demonstrates consistent CTSL activity despite pH variations. Then we conducted a series of experiments to investigate the impact of hyperglycemia on CTSL activity. Our findings showed that elevated glucose levels significantly stimulated both intracellular and extracellular CTSL activity in a dose-dependent manner in Huh7 cells (*Figure 4b and c*), which was consistent with our clinical data (*Figure 2a and h*). In contrast, insulin levels had no effect on CTSL activity in Huh7 cells (*Figure 4d and e*), indicating that it was hyperglycemia, rather than hyperinsulinemia, that boosted CTSL activity in diabetic patients.

We then evaluated CTSL activity in human and mouse biopsy tissues under high blood glucose condition. The demographic and clinical information for the human lung tissue samples donors in this study can be found in *Supplementary file 4*. Diabetic mice (*Lepr^db/db* mice) had significantly increased glucose, body weight and fasting insulin levels compared to control (*Lepr^db/+* mice) group (*Figure 4f–h*). We observed an elevation of CTSL activity in both diabetic mice and human lung tissues (*Figure 4i and j*), suggesting that high glucose condition may increase CTSL activity in the respiratory system in diabetic patients and mice in vivo. The increase of CTSL activity in diabetic mice liver tissue (*Figure 4i*) was consistent with the results of Huh7 hepatocytes (*Figure 4b*).

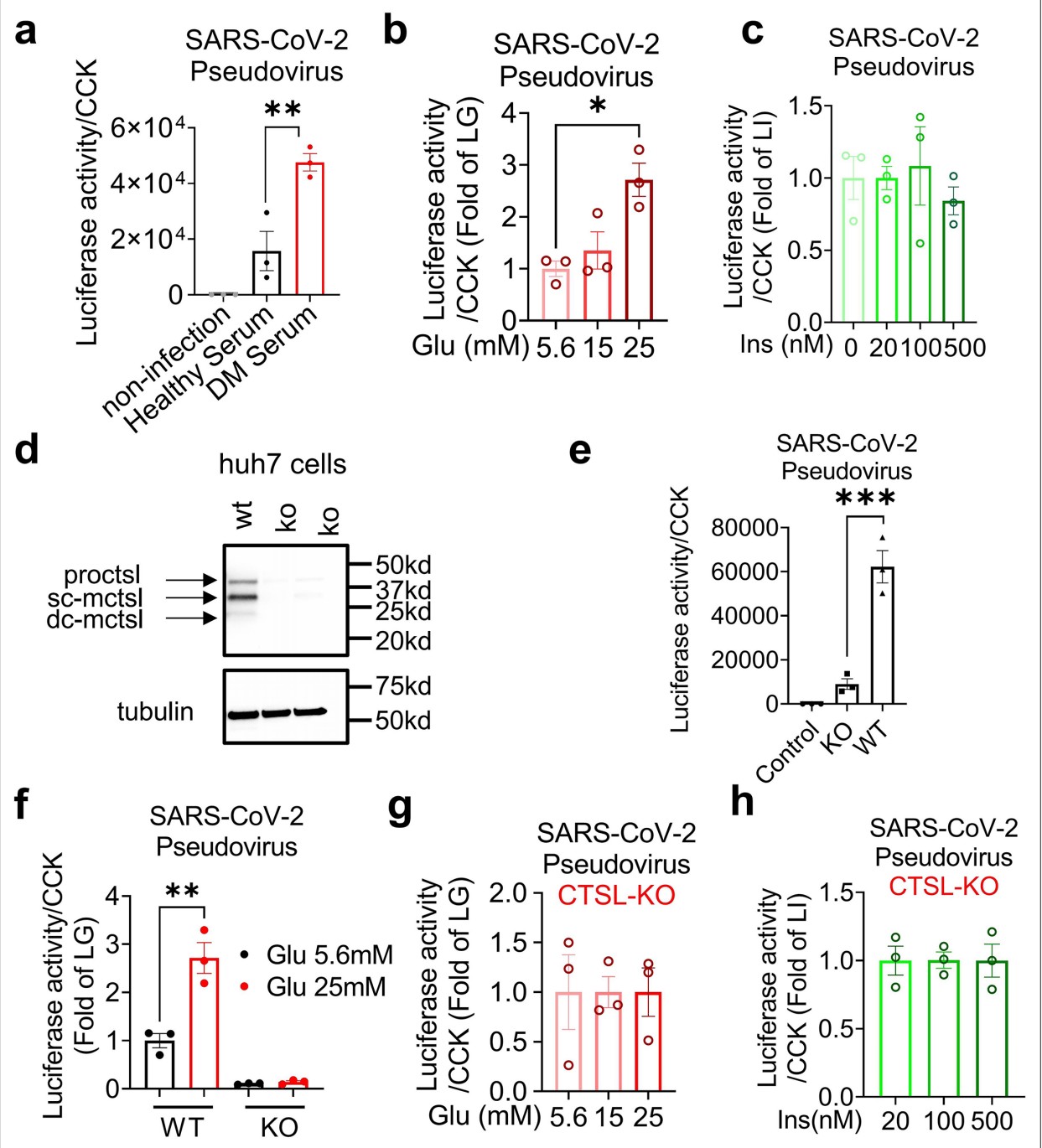

**Figure 3.** Hyperglycemia enhances SARS-CoV-2 infection through CTSL. Huh7 cells were infected with SARS-CoV-2 pseudovirus. (**a**) Wildtype (WT) cells cultured in sera from healthy and diabetic individuals were infected with SARS-CoV-2 pseudovirus ($1.3×10^4$ TCID$_{50}$/ml). Non-infected cells are used as control. The infection levels, as indicated by luciferase activities, were adjusted by cell viability, as indicated by CCK (n=3). (**b**) The SARS-CoV-2 infection rate of WT cells after being cultured in different doses of glucose (5.6 mM, 15 mM and 25 mM) (n=3). (**c**) The SARS-CoV-2 infection rate of WT cells after being cultured in different doses of insulin (0 nM, 20 nM, 100 nM and 500 nM) (n=3). (**d**) Comparison of CTSL expression in WT and *CTSL* knockout (KO) cell lines. (**e–h**) The SARS-CoV-2 pseudovirus infection of WT and *CTSL* KO cells. € The SARS-CoV-2 infection rate of control, *CTSL* KO, and WT cells (n=3). (**f**) The SARS-CoV-2 infection rate of WT and *CTSL* KO Huh7 cells cultured in different doses of glucose (5.6 mM and 25 mM) (n=3). (**g**) The SARS-CoV-2 infection rate of *CTSL* KO Huh7 cells cultured in different doses of glucose (5.6 mM, 15 mM and 25 mM) (n=3). (**h**) The SARS-CoV-2 infection rate of *CTSL* KO Huh7 cells cultured in different doses of insulin (20 nM, 100 nM and 500 nM) (n=3). Statistical significance was assessed by one-way ANOVA with Tukey's post hoc test (**a–c, e–h**). The data are presented as the means ± SEM. *p<0.05, **p<0.01, ***p<0.001.

The online version of this article includes the following source data for figure 3:

*Figure 3 continued on next page*

*Figure 3 continued*

**Source data 1.** The original data supporting *Figure 3*.

**Source data 2.** Raw western blots shown in *Figure 3*.

## High glucose levels stimulate CTSL maturation

To investigate the impact of high blood glucose on CTSL expression, we first measured the mRNA levels of CTSL under different glucose concentrations using D-glucose and L-glucose (*Figure 5—figure supplement 1*). While D-glucose is commonly used as a major energy source, L-glucose cannot be absorbed by cells but has similar physical properties to D-glucose, making it an ideal control. Surprisingly, our results showed that glucose or insulin levels did not affect CTSL mRNA levels in Huh7 cells, mouse tissues or human tissues (*Figure 5—figure supplement 1*), indicating that glucose did not influence CTSL transcription.

CTSL undergoes several forms during its translation and post-translational maturation process. The immature pro-cathepsin L (proCTSL, 39 kDa) possesses an N-terminal proregion, which acts as an autoinhibitor. The protein is translocated through the endoplasmic reticulum (ER)-Golgi apparatus-lysosome axis, and the proregion is removed in the acidic environment in lysosome results in either single chain mature cathepsin L (sc-mCTSL, 31 kDa) or double chain mature cathepsin L (dc-mCTSL, 24 kDa) (*Figure 5a*; *Coutinho et al., 2012*; *Ishidoh and Kominami, 2002*; *Reiser et al., 2010*). Only mature CTSL in lysosome has catalytic activity against specific substrates, while proCTSL does not (*Ishidoh and Kominami, 2002*).

Interestingly, we found that high glucose levels promoted CTSL maturation, whereas insulin had no effect on this process, as shown in *Figure 5b*. Our results indicated that high D-glucose levels reduced proCTSL and increased sc-mCTSL and dc-mCTSL in a glucose dose-dependent manner (*Figure 5b*, *Figure 5—figure supplement 2*). The results also confirmed that only D-glucose induced CTSL maturation from proCTSL to mCTSL, while L-glucose had no such effects (*Figure 5b and c*). Similar effects of high glucose on CTSL maturation were also observed in diabetic mice and human tissues compared with their healthy counterparts (*Figure 5d and e*). The differences observed in the processing of CTSL between cells (*Figure 5b*) and tissues (*Figure 5d–e*) may be attributed to the complexities inherent in tissue samples, which can impact the clarity of the images. Furthermore, in human tissue samples, it is pertinent to consider that patients in the diabetes group had their blood glucose levels controlled within or near the normal range prior to lung surgery. As a result, the evidence supporting CTSL maturation in human lung tissue blotting images may be less compelling. These results suggested that hyperglycemia rather than hyperinsulinemia or other physical parameters associated with CTSL maturation. This was consistent with our previous data that elevated glucose levels enhanced CTSL activity (*Figure 4b*), since only mature CTSL has enzymatic activity.

## High glucose facilitates CTSL maturation via translocation

We have shown that high glucose promoted CTSL maturation by converting proCTSL into mCTSL (*Figure 5*, *Figure 5—figure supplement 2*). However, the mechanism underlying this process remained unclear. As mentioned previously, CTSL maturation depends on structural activation via the ER-Golgi apparatus-lysosome axis and acid activation at low pH environment (*Figure 6a*; *Ishidoh and Kominami, 2002*; *Reiser et al., 2010*). Only the CTSL in lysosome is processed into mature form and has proteolytic activity, while the CTSL in the ER or Golgi apparatus is immature and does not have enzymatic activity. Based on this, we hypothesized that high blood glucose may drive CTSL maturation via the ER-Golgi apparatus-lysosome axis. We labeled specific proteins in each organelle: calreticulin for ER, GM130 for Golgi apparatus and lamp1 for lysosome (*Figure 6—figure supplement 1*). We also confirmed CTSL expression in Huh7 cells (*Figure 6—figure supplement 2*). Confocal microscopy results revealed that under low glucose conditions, CTSL tended to co-localize within the ER rather than lysosomes (*Figure 6b and e*). This finding suggests that CTSL primarily existed in an immature form under low glucose conditions. As glucose level increased, CTSL in the Golgi apparatus remained unchanged (*Figure 6c and f*), while co-localized largely increased in the lysosomes (*Figure 6d and g*), suggesting CTSL predominantly existed in mature form under high-glucose conditions. Therefore, we concluded that high glucose facilitated CTSL translocation through the ER-Golgi apparatus-lysosome axis.

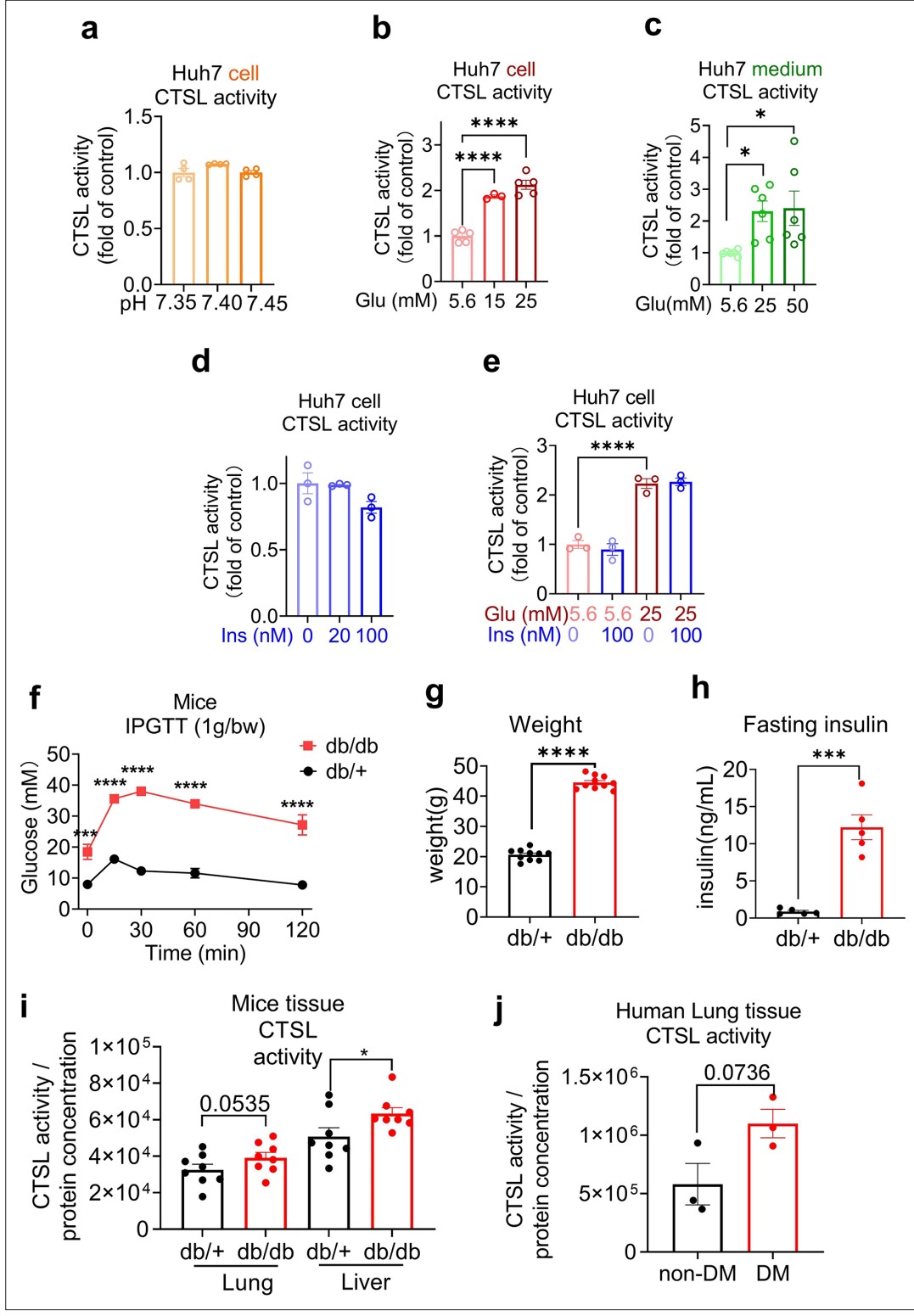

**Figure 4.** Elevation of glucose levels enhance CTSL activity. Effects of high glucose levels on CTSL activity in Huh7 cells, as well as in biopsy samples of mice and diabetic patients. (**a**) Intracellular CTSL activity was measured in Huh7 cells cultured in different pH as indicated (n=4). (**b**) Intracellular CTSL activity was measured in Huh7 cells cultured in different glucose concentrations as indicated (5.6 Glu group, n=5; 15 Glu group, n=3; 25 Glu group, n=5). (**c**) Extracellular CTSL activity was measured in Huh7 cells cultured in different glucose concentrations as indicated (n=6). (**d**) Intracellular CTSL activity was measured in Huh7 cells cultured in different insulin concentration

*Figure 4 continued on next page*

*Figure 4 continued*

as indicated (n=3). (**e**) Intracellular CTSL activity was measured in Huh7 cells cultured in different glucose and insulin concentrations as indicated (n=3). (**f**) Blood glucose levels during the intraperitoneal glucose tolerance test (IPGTT) in $Lepr^{db/db}$ mice and $Lepr^{db/+}$ mice (n=5). (**g**) Body weight of $Lepr^{db/db}$ mice and $Lepr^{db/+}$ mice was measured (n=10). (**h**) Fasting insulin levels were measured in $Lepr^{db/db}$ mice and $Lepr^{db/+}$ mice (n=5). (**i**) CTSL activity was measured in the lung and liver biopsy samples of $Lepr^{db/db}$ mice and $Lepr^{db/+}$ mice (n=8). (**j**) CTSL activity was measured in human lung biopsy samples from diabetic (DM) and non-diabetic patients (n=3). Statistical significance was assessed by one-way ANOVA with Tukey's post hoc test (**a–e**), two-way ANOVA with Sidak's multiple comparisons test (**f**) and unpaired *t*-test (**g–j**). The data are presented as the means ± SEM. *$p<0.05$, ***$p<0.001$, ****$p<0.0001$.

The online version of this article includes the following source data for figure 4:

**Source data 1.** The original data supporting *Figure 4*.

## Discussion

Almost immediately after the SARS-CoV-2 emerged, it became apparent that individuals with chronic conditions, including diabetes, were disproportionately affected, with a heightened risk of hospitalization and mortality (*Williamson et al., 2020*). Diabetes may increase susceptibility to severe SARS-CoV-2 infections for various suggested reasons. These include higher viral titer, relatively low functioning T lymphocytes that lead to decreased viral clearance, vulnerability to hyperinflammation and cytokine storm syndrome, and comorbidities associated with type 2 diabetes, such as cardiovascular disease, non-alcoholic fatty liver disease, hypertension, and obesity (*Mazucanti and Egan, 2020*; *Moradi-Marjaneh et al., 2021*). Additionally, other risk factors that may contribute to the severity of infection include increased expression of angiotensin-converting enzyme 2 (ACE2) (*Rao et al., 2020*) and furin (*Fernandez et al., 2018*). However, most current studies on COVID-19 and diabetes focus on epidemiological evidence and biomarker features, but few investigate the causal link and underlying mechanisms of how hyperglycemia enhances SARS-CoV-2 infection, confirmed by human body fluids, biopsies, and animal models. The underlying mechanisms by which diabetes or hyperglycemia exacerbates COVID-19 remain to be fully elucidated.

This study identified that CTSL maturation induced by hyperglycemia may contribute to the higher mortality and severity of COVID-19 in patients with diabetes. While our initial study involved 62 COVID-19 patients, with 31 having diabetes and 31 without, matching based on gender and age, we acknowledged the challenge of obtaining balanced gender distribution in both groups due to the difficulty of collecting blood samples from COVID-19 patients. To mitigate potential gender bias resulting from small sample sizes, we conducted a supplementary clinical study involving 122 non-COVID-19 volunteers, including 61 individuals with diabetes and 61 without. The percentage of males in the diabetes group was 50.8%, while in the healthy group, males constituted 44.3% (p-value = 0.468), indicating no significant gender bias. The clinical data from this study indicated that human plasma CTSL activity and concentration were correlated with acute (in euglycemic participants under high-glucose clamp conditions) and chronic (in diabetic patients, both with and without COVID-19) hyperglycemia, respectively.

Using lung tissue samples from diabetic and non-diabetic patients, as well as $Lepr^{db/db}$ mice and $Lepr^{db/+}$ mice, we found that diabetic conditions increased CTSL activity in both humans and mice. The liver is a significant target organ for COVID-19 (*Gupta et al., 2020*). Despite potential limitations, such as generalization of liver function abnormalities and lack of tissue specificity in SARS-CoV-2 impact, Huh7 cells offer practical advantages as a mature cell model for studying SARS-CoV-2 infection, including accessibility, susceptibility to infection, and stable proliferation (*Nie et al., 2021*; *Ciotti et al., 2020*). Taking all these factors into consideration, we have ultimately chosen to utilize the hepatoma cell line to investigate how hyperglycemia induces CTSL maturation and subsequently promotes SARS-CoV-2 infection. High glucose promoted SARS-CoV-2 infection in WT cells, while *CTSL* KO cells showed reduced susceptibility to high glucose promoting effects. Mechanistically, we proposed that hyperglycemia promoted CTSL maturation by accelerating its translocation from the ER to lysosome via Golgi apparatus. This condition increased the functionality of CTSL, which cleaved the spike protein of SARS-CoV-2, promoting virus membrane fusion and infection (*Figure 6h*). In our study, the term 'diabetes' encompasses the condition of hyperglycemia in a broad sense, rather than specifically indicating type 1 diabetes (T1DM) or type 2 diabetes (T2DM). This broader definition

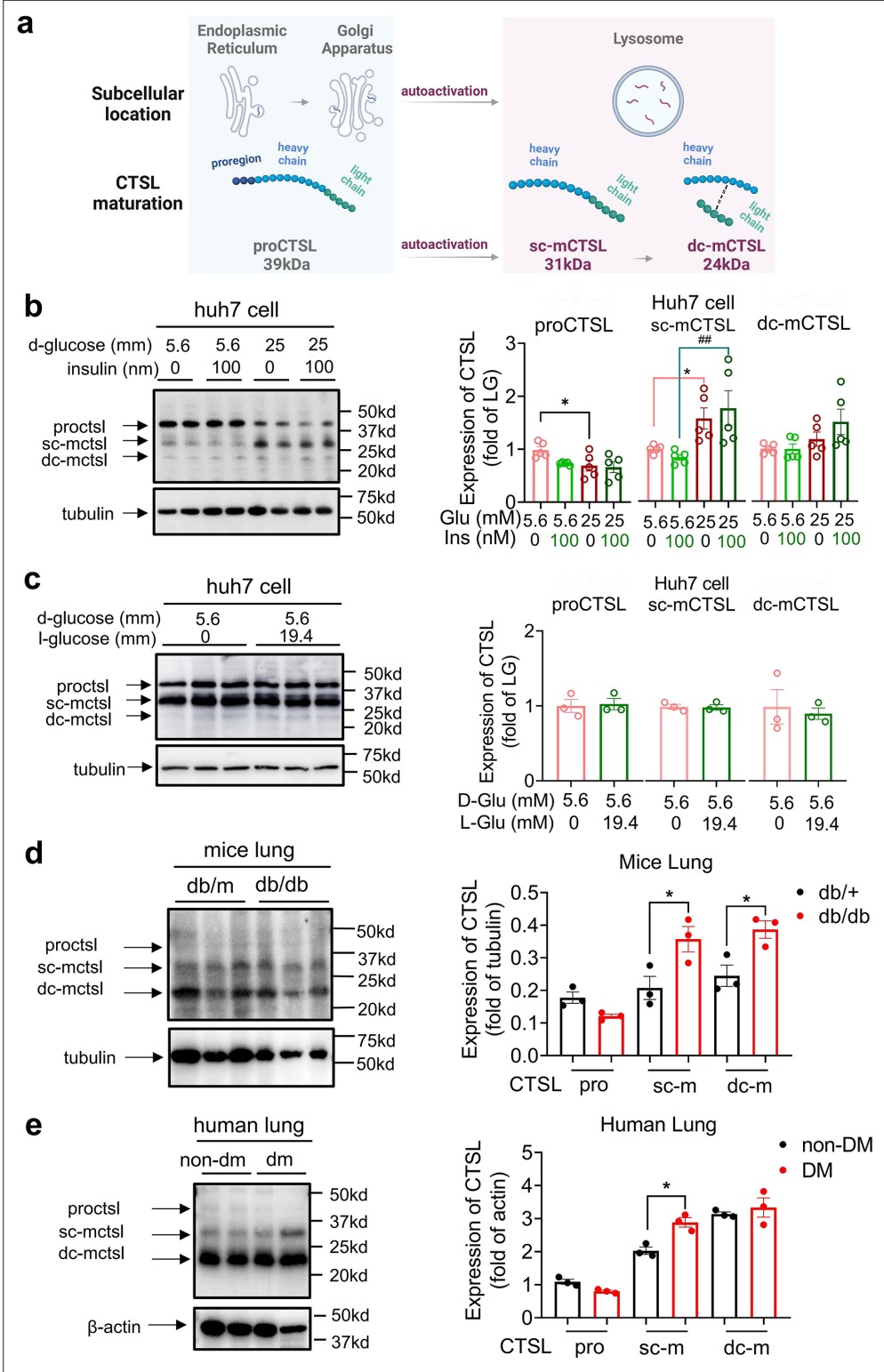

**Figure 5.** High glucose levels stimulate CTSL maturation. (**a**) Schematic of the CTSL maturation process. Pro-cathepsin L (ProCTSL, 39 kDa) in endoplasmic reticulum (ER) and Golgi apparatus translocated to the lysosome and autoactivated into the single chain mature cathepsin L (sc-mCTSL, 31 kDa) and double chain mature cathepsin L (dc-mCTSL, 24 KDa). (**b**) Western blot analysis of CTSL protein in Huh7 cells cultured with different doses of D-glucose and insulin as indicated (n=5). (**c**) Western blot analysis of CTSL protein in Huh7 cells cultured with 5.6 mM D-glucose or D-glucose plus 19.4 mM L-glucose as indicated (n=3). (**d**) Western blot analysis of CTSL protein in

*Figure 5 continued on next page*

*Figure 5 continued*

lung tissues from *Lepr^db/db* mice and *Lepr^db/+* mice (n=3). (**e**) Western blot analysis of CTSL protein in human lung tissues from non-diabetic and diabetic patients (n=3). Statistical significance was assessed by one-way ANOVA with Tukey's post hoc test (**b–e**). The data are presented as the means ± SEM. *$p<0.05$, ##$p<0.01$.

The online version of this article includes the following source data and figure supplement(s) for figure 5:

**Source data 1.** The original data supporting *Figure 5*.

**Source data 2.** Raw western blots shown in *Figure 5*.

**Figure supplement 1.** CTSL mRNA levels remain unchanged under different glucose conditions.

**Figure supplement 1—source data 1.** The original data supporting *Figure 5—figure supplement 1*.

**Figure supplement 2.** CTSL protein expression in Huh7 cells under different D-glucose concentrations.

**Figure supplement 2—source data 1.** The original data supporting *Figure 5—figure supplement 2*.

**Figure supplement 2—source data 2.** Raw western blots shown in *Figure 5—figure supplement 2*.

---

aligns with the scope of our research objectives and findings, particularly observed in the cell experiments conducted.

*Figure 2h–j* illustrate the impact of acute hyperglycemia on CTSL concentration and activity in 15 healthy male volunteers over a 160 min period. During this short timeframe, CTSL concentration remained stable, as evidenced by consistent RNA results from cells exposed to varying glucose levels (*Figure 5— figure supplement 1*). However, there was a significant increase in CTSL activity, indicating that glucose elevation rapidly triggers CTSL maturation through propeptide cleavage. This activation process occurs more rapidly than CTSL protein synthesis. In summary, acute hyperglycemia specifically elevates CTSL activity, while chronic hyperglycemia may impact both CTSL activity and concentration (*Figure 2a–d*). To date, there have been limited studies investigating the relationship between cathepsin maturation and glucose. In 1998, Tournu C, et al. reported that D-glucose did not impact mRNA levels for CTSB or CTSL or secretion of proCTSL. However, D-glucose did significantly enhance the amount of mature forms of CTSB and CTSL (*Tournu et al., 1998*). More recently, Shi Q, et al. found that increased glucose metabolism promotes O-GlcNAcylation of the lysosome-encapsulated protease CTSB, leading to elevated levels of mature CTSB in macrophages and secretion in the tumor microenvironment (*Shi et al., 2022*). These findings support our evidence that hyperglycemia drives CTSL maturation.

ACE2 has previously been identified as a critical host cell surface receptor that enables SARS-CoV-2 entry into host cells (*Wrapp et al., 2020*). While some studies have reported that glucose can increase ACE2 expression in cell lines (*Härdtner et al., 2013*), numerous other studies have found that ACE2 is downregulated in diabetic patients (*Mizuiri et al., 2008*; *Reich et al., 2008*). Garreta et al. recently conducted a study using a human kidney organoid system to investigate the impact of diabetes on SARS-CoV-2 infections. The study revealed that hyperglycemia enhanced SARS-CoV-2 infection and hyperglycemic human kidney organoids had elevated ACE2 levels (*Garreta et al., 2022*). Therefore, it remains controversial whether diabetes results in up- or downregulation of ACE2. In our study, we evaluated plasma levels of ACE2, CTSL, and CTSB in COVID-19 patients with and without diabetes. We found that only CTSL levels were significantly increased in diabetic patients compared to non-diabetic patients and varied during the course of COVID-19.

In addition to CTSL, there may be other bioactive factors involved in mediating SARS-CoV-2 infection in patients with diabetes. A recent study revealed that diabetic patients have lower levels of serum 1,5-anhydro-D-glucitol (1,5-AG), a small-molecule metabolite in human blood that exhibits potent antiviral activity against SARS-CoV-2. The reduced levels of 1,5-AG have been associated with increased viral loads and severe respiratory tissue damage caused by SARS-CoV-2. Mechanistically, the study found that 1,5-AG binds directly to the S2 subunit of the spike protein, which disrupts virus-host membrane fusion and inhibits infection (*Tong et al., 2022*). Therefore, we propose that diabetes may promote COVID-19 infection through multiple factors, and CTSL is only one of several important factors.

Apart from diabetes, other comorbidities such as hypertension and CHD are also prevalent in COVID-19 patients. Interestingly, our study revealed a strong correlation between CTSL activity and concentration with hypertension and CHD in these patients. Using an angiotensin II-induced hypertension model, researchers observed an increase in blood pressure and CTSL activity (*Lu et al., 2020*). Whether these chronic comorbidities contribute to increased morbidity and mortality of COVID-19 by increasing CTSL activity and concentration requires further investigation in the future.

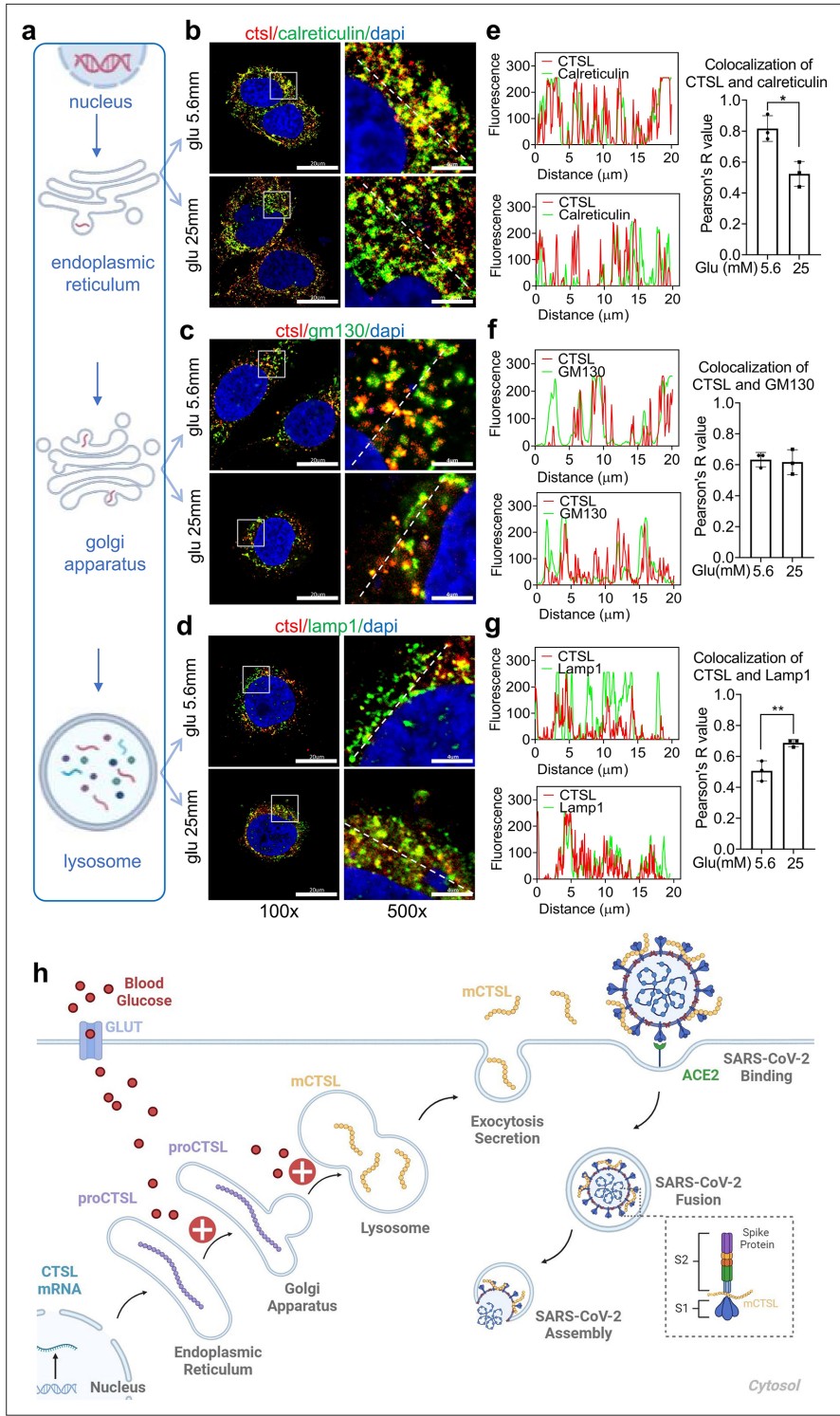

**Figure 6.** High glucose promotes CTSL translocation from endoplasmic reticulum to lysosome and enhances SARS-CoV-2 infection. (**a**) A diagram illustrating the process of CTSL translocation via the endoplasmic reticulum (ER)-Golgi-lysosome axis. (**b–d**) Immunofluorescent staining of Huh7 cells cultured in 5.6 mM or 25 mM glucose as indicated. Co-localization analysis of CTSL (labeled red) and different organelles (labeled green) was performed using CTSL and organelle marker protein antibodies. (**b**) calreticulin for ER, (**c**) GM130 for Golgi apparatus, and (**d**) lamp1 for lysosome. (**e–g**) Fluorescence co-localization intensity analysis of the dashed line on the 500 times enlarged immunofluorescence picture. Fluorescence co-localization intensity was calculated using the Plot Profile tool in Image J software (n=3). Scale bars, 20 µm for 100 x and 4 µm for 500 x. (**h**) Proposed mechanisms

*Figure 6 continued on next page*

*Figure 6 continued*

of hyperglycemia drives CTSL maturation and enhances SARS-CoV-2 infection. (1) Blood glucose increased in diabetic patients. (2) Hyperglycemia promoted CTSL maturation through the ER-Golgi-lysosome axis. (3) CTSL activity increased and facilitated SARS-CoV-2 entry, by cleaving the spike protein (consist of S1 and S2 subunits), and enhanced COVID-19 severity in diabetic patients. All statistical significance was assessed using unpaired *t*-test. The data are presented as the means ± SEM. *p<0.05, **p<0.01.

The online version of this article includes the following source data and figure supplement(s) for figure 6:

**Source data 1.** The original data supporting *Figure 6*.

**Figure supplement 1.** Immunofluorescent staining of organelle markers representing the endoplasmic reticulum (ER), Golgi apparatus and lysosome.

**Figure supplement 2.** Immunofluorescent staining of CTSL in Huh7 cells.

In conclusion, our study demonstrates that hyperglycemia drives the maturation and activation of CTSL, for only mature form of CTSL gains its function of proteolysis. Therefore, targeting CTSL may be a promising therapeutic strategy for diabetic comorbidities and complications (*Li et al., 2022*).

# Materials and methods

**Key resources table**

| Reagent type (species) or resource | Designation | Source or reference | Identifiers | Additional information |
|---|---|---|---|---|
| Strain, strain background (*Mus musculus*) | Lepr$^{db/db}$ mice | Vital River Laboratories | N/A | N/A |
| Strain, strain background (*Mus musculus*) | Lepr$^{db/+}$ mice | Vital River Laboratories | N/A | N/A |
| Cell line (*Homo sapiens*, liver) | The Huh7 cell line | Cell Resource Center, Chinese Academy of Medical Sciences | N/A | N/A |
| Biological sample (*Homo sapiens*) | Blood samples from patients with COVID-19 | Beijing Youan Hospital, Capital Medical University; Wuhan Union Hospital, Huazhong University of Science and Technology | N/A | N/A |
| Biological sample (*Homo sapiens*) | Blood samples from volunteers without COVID-19 | Beijing Tongren Hospital, Capital Medical University | N/A | N/A |
| Biological sample (*Homo sapiens*) | Human lung samples | Beijing Youan Hospital, Capital Medical University | N/A | N/A |
| Antibody | CTSL (Goat Polyclonal) | R&D System | AF952; RRID:AB_355737 | Dilution (1:2000) |
| Antibody | α-tubulin (Mouse Monoclonal) | Proteintech | 66031–1-Ig; RRID:AB_11042766 | Dilution (1:1000) |
| Antibody | β-actin (Mouse Monoclonal) | Proteintech | 66009–1-Ig; RRID:AB_2687938 | Dilution (1:1000) |
| Antibody | Calreticulin (Rabbit Monoclonal) | Cell Signaling Technology | 12238; RRID:AB_2688013 | Dilution (1:200) |
| Antibody | GM130 (Mouse) | BD Biosciences | 610822; RRID:AB_398141 | Dilution (1:200) |
| Antibody | Lamp1 (Mouse Monoclonal) | PTM BIO | PTM-5775 | Dilution (1:200) |
| Antibody | Goat IgG | Beyotime | A7007 | Dilution (1:500) |
| Antibody | Rabbit IgG | Beyotime | A7016; RRID:AB_2905533 | Dilution (1:500) |

*Continued on next page*

*Continued*

| Reagent type (species) or resource | Designation | Source or reference | Identifiers | Additional information |
|---|---|---|---|---|
| Antibody | Mouse IgG | Beyotime | A7028; RRID:AB_2909433 | Dilution (1:500) |
| Sequence-based reagent | Primer: CTSL (human) - F | This paper | AAACTGGGAGGCTTATCTCACT | N/A |
| Sequence-based reagent | Primer: CTSL (human) - R | This paper | GCATAATCCATTAGGCCACCAT | N/A |
| Sequence-based reagent | Primer: Ctsl (mouse) - F | This paper | CTACACAACGGGGAATACAGC | N/A |
| Sequence-based reagent | Primer: Ctsl (mouse) - R | This paper | CATTGGTCATGTCACCGAAGG | N/A |
| Sequence-based reagent | Primer: ACTB (human) – F | This paper | TCATGAAGTGTGACGTGGACATC | N/A |
| Sequence-based reagent | Primer: ACTB (human) – R | This paper | CAGGAGGAGCAATGATCTTGATCT | N/A |
| Sequence-based reagent | Primer: Actb (mouse) – F | This paper | GTGACGTTGACATCCGTAAAGA | N/A |
| Sequence-based reagent | Primer: Actb (mouse) – R | This paper | GCCGGACTCATCGTACTCC | N/A |
| Commercial assay or kit | Human ACE2 Elisa kit | Cloud-Clone Corp | Cat. No L220216094 | N/A |
| Commercial assay or kit | Human CTSL ELISA Kit | Elabscience | Cat. No E-EL-H0671 | N/A |
| Commercial assay or kit | Human CTSB ELISA kit | Elabscience | Cat. No E-EL-H6151 | N/A |
| Commercial assay or kit | Mouse insulin ELISA kit | Millipore | Cat. No EZRMI-13K; RRID:AB_2783856 | N/A |
| Commercial assay or kit | Britelite Plus Kit | Perkinelmer | Cat. No 6066761 | N/A |
| Commercial assay or kit | Cell Counting Kit | Transgen | Cat. No FC101-04 | N/A |
| Commercial assay or kit | RNAprep Pure Cell/Bacteria kit | Tiangen | Cat. No DP430 | N/A |
| Commercial assay or kit | RNAprep Pure Tissue kit | Tiangen | Cat. No DP431 | N/A |
| Chemical compound, drug | Ac-FR-AFC | R&D | Cat. No ES009 | N/A |
| Chemical compound, drug | TransScript First-Strand cDNA Synthesis SuperMix | Transgen | Cat. No AT301 | N/A |
| Chemical compound, drug | TransStart Tip Green qPCR SuperMix | Transgen | Cat. No AQ141 | N/A |
| Software, algorithm | Image J | Image J Software | https://imagej.net/software/imagej/ | N/A |
| Software, algorithm | Graphpad prism 7.0 | GraphPad Software | https://www.graphpad.com | N/A |
| Software, algorithm | SPSS for Windows 17.0 | IBM SPSS Software | https://www.ibm.com/analytics/spss-statistics-software | N/A |
| Software, algorithm | Microsoft Office Home and Student 2019 | Microsoft Corporation | https://www.microsoft.com/microsoft-365 | N/A |

## Experimental model and study participant details

### Patients and clinical samples

The study in patients was approved by the Ethics Committee of Beijing Tongren Hospital, Capital Medical University (TRECKY2020-013, TRECKY2021-202). The retrospective cohort study included 207 COVID-19 patients from two hospitals. 120 adult COVID-19 inpatients admitted to Wuhan Union Hospital, Huazhong University of Science and Technology (Wuhan, China) between January 29 and March 20, 2020. Another 87 consecutive COVID-19 inpatients were hospitalized at Beijing Youan Hospital, Capital Medical University (Beijing, China) between January 21 and April 30, 2020. Thirty-one COVID-19 patients with diabetes and 31 COVID-19 patients without diabetes were matched for gender and age and included in the final analysis. The clinical features are presented in *Supplementary file 1*. SARS-CoV-2 was detected in respiratory specimens using real-time RT-PCR, following the protocol recommended by the World Health Organization. COVID-19 was classified into four categories: mild, moderate, severe and critical, according to the clinical classification criteria (http://www.nhc.gov.cn/). Patients from Beijing Youan Hospital, Capital Medical University were further followed up. They experienced a mean of 14 days of hospitalization and were followed up on the 14th day (Day 14) and 28th day (Day 28) after discharge from the hospital. Blood samples were collected shortly after the admission to the hospital (Day 0) and on Day 14 and Day 28. Demographic, clinical, and laboratory data were extracted from the electronic hospital information system using a standardized form.

Another total of 122 age- and gender-matched diabetic and non-diabetic volunteers without COVID-19 were recruited in Beijing Tongren Hospital, Capital Medical University. Blood samples were collected after overnight fasting for the determination of CTSL activity and concentration and other biochemical parameters. All biochemical measurements have participated in the Chinese Ministry of Health Quality Assessment Program. The demographic and clinical characteristics are shown in *Supplementary file 2*.

Human lung tissue samples were obtained from six patients who underwent lung surgery at Beijing Tongren Hospital between March 22 and June 22, 2022. The baseline characteristics are presented in *Supplementary file 4*.

The plasma samples of hyperglycemic clamp study were from a previously conducted clinical trial (NCT03972215). Fifteen healthy male research subjects were received a 160 min hyperglycemic clamp study with a baseline blood glucose level +6.9 mmol/L as the target level. Blood samples were obtained at intervals throughout the clamp study. The plasma was collected and stored at –80 °C until use.

### Experimental mice

The study in mice was approved by the Ethics Committee of Beijing Tongren Hospital, Capital Medical University (TRLAWEC2023-S194). The study used 10-week-old *Lepr*$^{db/db}$ mice as diabetic mode and *Lepr*$^{db/+}$ mice as their healthy control, maintained on a KBS background. All mice were obtained from Vital River Laboratories (Beijing, China). The mice were housed at constant humidity and temperature, with a 12 hr light/dark cycle. The protocols for the use of mice were approved by the Ethical Review Committee at the Institute of Zoology, Capital Medical University.

### Cell lines and reagents

The Huh7 (*Homo sapiens*, liver) cell line (Cell Resource Center, Chinese Academy of Medical Sciences, Beijing, China), was maintained in high glucose Dulbecco's modified Eagle's medium (DMEM; Sigma-Aldrich, St. Louis, MO, USA) supplemented with streptomycin (100 mg/ml), penicillin (100 units/ml), and fetal bovine serum (10%, Gibco, Carlsbad, CA). The cells were maintained at 37 °C in a humidified atmosphere of 5% $CO_2$ and 95% air. The cell lines were cultured with an anti-mycoplasma drug, and upon visual inspection, no mycoplasma contamination was observed.

## Method details

### Detection of SARS-CoV-2 entry related host biomarkers

Plasma samples of patients with COVID-19 at Day 0, Day 14, and Day 28 were collected and stored at –80 °C within 2 hr. The samples were analyzed using commercially available enzyme-linked immunosorbent assays (ELISA) following the manufacturer's instructions. All samples were detected without

virus inactivation to retain the original results in a P2 +biosafety laboratory. ACE2 was measured using the Human ACE2 Elisa kit (Cloud-Clone Corp, Cat. No L220216094). CTSL and CTSB were measured using the Human CTSL ELISA Kit (Elabscience, Cat. No E-EL-H0671) and Human CTSB ELISA kit (Elabscience, Cat. No E-EL-H6151). The kits were designed for usage with human serum or plasma samples and showed no cross-reactions.

## Production of pseudovirus

The SARS-CoV-2 pseudovirus were generated with the incorporation of SARS-CoV-2 spike protein (SARS-2-S) into vesicular stomatitis virus (VSV)-based pseudovirus system. The pseudoviruses used in the current study have been validated in previous studies (*Lv et al., 2020*; *Whitt, 2010*). For this VSV-based pseudovirus system, the backbone was provided by VSV-G pseudotyped virus (G*ΔG-VSV) that packages expression cassettes for firefly luciferase instead of VSV-G in the VSV genome (*Nie et al., 2020*). Therefore, the luciferase activity of VSV phosphoprotein (VSV-P) were used for indicators of pseudovirus infection.

## Pseudovirus infection in vitro

Huh7 cells were plated in 96-well plates and allowed to adhere until they reached 70% confluency. Subsequently, these cells were cultured with different medium or serum obtained from diabetic or euglycemic individuals as indicated. Following this, the cells were infected with SARS-CoV-2 pseudo-virus at $1.3 \times 10^4$ TCID$_{50}$/ml at 37 °C. After coincubation with pseudovirus of 24 hr, the activities of firefly luciferases were measured on cell lysates using luciferase substrate, Britelite Plus Kit (Perkinelmer, Cat. No 6066761) according to the manufacturer's instructions. The firefly luciferase activity was measured rapidly using a luminometer (Turner BioSystems, USA) as described previously (*Yang et al., 2017*). Cell viability were measured by CCK assay using Cell Counting Kit (Transgen, Cat. No FC101-04). The infection rates were adjusted by cell viability.

## Establishment of *CTSL*-KO Huh7 cell line via CRISPR/Cas9

To produce the *CTSL*-KO cell line, we utilized CRISPR/Cas9 technology. The single guide RNA (sgRNA) was designed using the Zhang Lab Guide Design Resources (https://zlab.bio/guide-design-resources) tool. The sgRNA scaffold was commercially obtained from Sangon, with the sequence designed as 5'-ctttgtggacatccctaagc-3'. For sgRNA and Cas9 protein to enter into Huh7 cell, electroporation-mediated transfection was performed. First, $4 \times 10^5$ Huh7 cells were centrifuged and re-suspended in 10 μL of Buffer R following the manufacturer's instructions (Invitrogen, USA) for electroporation. Next, Cas9 protein (1 μg) and sgRNA (0.2 μg) were added to each sample and mixed gently (Cas9 protein: sgRNA at a 1:1 molar ratio). Huh7 cells were electroporated for 5 times to minimize the *CTSL* gene on a Neon Transfection device (Invitrogen, USA).

## Generation of *CTSL*-KO monoclonal Huh7 cell line

Cells were isolated from the stable *CTSL*-KO Huh7 cell pool by trypsinization and any cell clumps were broken up into individual cells. Cells concentration was quantitated in this homogenized cell solution with a cell counter. Then, 100 μL of the 5 cells/mL solution was transferred into each well of a 96-well plate. By doing this, the average density of 0.5 cells/well of the plate was seeded. Seeding an average of 0.5 cells/well ensured that some wells received a single cell, while minimizing the likelihood that any well receives more than one cell. Then we observed and recorded the cell growth in the plate for the following 30 days. Once the cells have expanded but before they become over-confluent, we trypsinized the cells and expanded them to larger culture dishes.

## RNA extraction and quantitative real-time PCR analyses

Total RNA was extracted and purified from the cultured Huh7 cells, human and mouse tissues using RNAprep Pure Cell/ Bacteria kit (Tiangen, Cat. No DP430) and RNAprep Pure Tissue kit (Tiangen, Cat. No DP431) according to the manufacturer's instructions. RNA (0.5 μg) was reverse transcribed to cDNA in a final volume (20 μL) using TransScript First-Strand cDNA Synthesis SuperMix (Transgen, Cat. No AT301). RT-PCR analyses were performed with TransStart Tip Green qPCR SuperMix (Transgen, Cat.

No AQ141). Gene expression values were normalized to the control (β-actin) level. Quantitative real-time PCR (qRT-PCR) and data collection were done on a LightCycler 96 system (Roche, Switzerland).

## Intraperitoneal glucose tolerance test (IPGTT)

The $Lepr^{db/db}$ mice and $Lepr^{db/+}$ mice were subjected to an overnight fast lasting 16 hr, during which they were permitted unrestricted access to water. Subsequently, they received an intraperitoneal injection of 1 g/kg body weight glucose. Blood samples were obtained at 0, 15-, 30-, 60-, and 120 min post-glucose injection. Blood glucose levels were determined using an automatic glucometer (One Touch, LifeScan, USA), while insulin concentrations were evaluated using a highly sensitive mouse insulin ELISA kit (Millipore, Cat. No EZRMI-13K), according to the manufacturer's instructions.

## Western blot analysis

Total protein was extracted from Huh7 cells and human and mouse tissues. The protein amount was assessed using the BCA protein assay kit (Thermo, Cat. No WH333441). Samples of 30–50 µg of protein were separated by SDS-PAGE, transferred to PVDF membrane (Millipore, Cat. No 0000167358), and detected using enhanced chemiluminescent reaction (*Zhao et al., 2021a*).

## Analysis of CTSL activity

The activity of CTSL in plasma, human and mouse tissues, Huh7 cells and cell medium was evaluated using its specific substrate, Ac-FR-AFC (R&D, Cat. No ES009). Prior to measurement, the cell medium was concentrated using the ultra-tube (Milipore, Cat. No UFC501096). The test samples were evaluated in the presence of a reaction buffer (100 mM NaAc, 5 mM EDTA, pH 5.3). The reaction was conducted in 100 µL system (50 µL sample containing CTSL protein +47 µL reaction buffer +1 µL DTT (1 mM)+2 µL substrate Ac-FR-AFC (10 mM)) in 96-well black plates. The plated was cultured at 37 °C for 2 hr in light avoidance incubator. The fluorescence emitted from the samples was then measured using a fluorescence plate reader (Infinite 200, TECAN, China) at the Ex = 380 nm, Em = 460 nm wavelengths.

## Immunofluorescence assay

The CTSL, calreticulin, Golgi membrane protein 130 (GM130) and lysosomal associated membrane protein 1 (Lamp 1) distributions in the Huh7 cells were visualized by immunofluorescent staining. The CTSL, calreticulin, GM130 and Lamp1 antibody species source IgG were used as negative control (*Figure 6—figure supplement 1*). Briefly, Huh7 cells was cultured in 35 mm confocal dishes after poly-l-lysine coating. After high/low glucose treatment for 96 hr, Huh7 cells were then fixed by 4% paraformaldehyde (PFA) and permeabilized by a detergent 0.25% triton X-100. A specific primary antibody is applied on the Huh7 cell surface at 4 °C overnight. After wash out, the secondary antibody is applied at room temperature for 1 hr avoid from light. All pictures were captured under Laser Scanning Confocal Microscopy (FV3000RS, Olympus, Japan).

## Quantification and statistical analysis

Clinical data are shown as percentage or median, as appropriate. Comparison of continuous data between two independent groups was performed using the Mann–Whitney *U*-test. An unpaired *t* test was used for comparing the averages/means of two independent or unrelated groups. A paired t-test was used to test whether the mean difference between pairs of measurements is existing. Analysis of variance (ANOVA) was used for checking if the means of two or more categories are significantly different from each other. Spearman's rho test (two-tailed) was used to analyze nonparametric correlations of parameters correlated with CTSL levels and diabetes. Fluorescence intensity was calculated by Plot Profile tool in Image J software. Graphpad prism 7.0 software and SPSS for Windows 17.0 were used for statistical analysis, with statistical significance set at two-sided. *p<0.05, **p<0.01, ***p<0.001, ****p<0.0001.

## Materials Availability

All data generated or analyzed during this study are included in the manuscript and supporting files. Source data files have been provided for *Figures 1–6* and *Supplementary files 1-4*.

## Acknowledgements

We thank participants and staff of the case-control studies for their valuable contributions. This work was supported by grants from National Natural Science Foundation of China (82341076; 81930019) to JKY. This work was also supported by grants from National Natural Science Foundation of China (82300917), Beijing Municipal Administration of Hospitals Incubating Program (PX20240203) to MMZ.

## Additional information

### Funding

| Funder | Grant reference number | Author |
| --- | --- | --- |
| National Natural Science Foundation of China | 82341076 | Jin-Kui Yang |
| National Natural Science Foundation of China | 81930019 | Jin-Kui Yang |
| National Natural Science Foundation of China | 82300917 | Miao-Miao Zhao |
| Beijing Municipal Administration of Hospitals | PX20240203 | Miao-Miao Zhao |

The funders had no role in study design, data collection and interpretation, or the decision to submit the work for publication.

### Author contributions

Qiong He, Software, Methodology, Writing - original draft; Miao-Miao Zhao, Software, Formal analysis, Funding acquisition, Writing – review and editing; Ming-Jia Li, Xiao-Ya Li, Validation, Methodology; Jian-Min Jin, Ying-Mei Feng, Fangyuan Yang, Data curation; Li Zhang, Wei Jin Huang, Methodology; Jin-Kui Yang, Conceptualization, Resources, Supervision, Funding acquisition, Writing – review and editing

### Author ORCIDs

Miao-Miao Zhao ![ORCID] http://orcid.org/0000-0002-4671-2463
Jin-Kui Yang ![ORCID] https://orcid.org/0000-0002-5430-2149

### Ethics

Clinical trial registration TRECKY2020-013, TRECKY2021-202.
The clinical study protocol was approved by the Ethics Committee of Beijing Tongren Hospital, Capital Medical University.
The study was performed according to animal experimentation ethics (TRLAWEC2023-S194).

Reviewer #1 (Public review): https://doi.org/10.7554/eLife.92826.3.sa1
Reviewer #2 (Public review): https://doi.org/10.7554/eLife.92826.3.sa2
Author response https://doi.org/10.7554/eLife.92826.3.sa3

## Additional files

### Supplementary files

• Supplementary file 1. Comparison of clinical features of COVID-19 patients. Data are median (IQR) or n (%). $P$ values were calculated by Mann-Whitney $U$-test (†) or $\chi^2$ test (#), as appropriate for group comparison analyses.

• Supplementary file 2. Demographic and clinical characteristics of non-COVID-19 patients. Data are median (IQR) or n (%). $P$ values were calculated by Mann-Whitney $U$-test (†) or $\chi^2$ test (#), as appropriate for group comparison analyses.

• Supplementary file 3. Nonparametric correlations of parameters correlated with CTSL levels and

diabetes in non-COVID-19 individuals. Data are correlation coefficient (*P* value). Spearman's rho test (two-tailed).

• Supplementary file 4. Demographic and clinical characteristics of human lung tissues donor. Six enrolled patients undergoing lung surgery in general surgery department of Beijing Tongren Hospital ranging from March 22 to June 22, 2022.

• MDAR checklist

## Data availability

All data generated or analyzed during this study are included in the manuscript and supporting files. Source data files have been provided for Figures 1–6 and Supplementary files 1–3.

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
