## [Editor Report · eLife assessment]

This **valuable** study advances our understanding of why diabetes is a risk factor for more severe Covid-19 disease. The authors offer **convincing** evidence that cathepsin L is more active in diabetic individuals because of the presence of high glucose, where the main mechanism is increased cathepsin L maturation. This study should be of interest to researchers in diabetes, virology and immunology.

---

## [Referee Report · Reviewer #1 (Public review)]

Summary:

The study by He et al. investigates the relationship of an increased susceptibility of diabetes patients towards COVID-19. The paper raises the possibility that hyperglycemia-induced cathepsin L maturation could be one of the driving forces in this pathology, suggesting that an increased activity of CTSL leads to accelerated virus infection rates due to an elevated processing of the SARS-CoV-2 spike protein.

In a clinical case-control study, the team found that severity of corona infections was higher in diabetic patients, and their CTSL levels correlated well with the progression of the disease. They further showed an increase in CTSL activity in long term as well as acute hyperglycemia. SARS-CoV-2 increasingly infected cells that were cultured in serum from diabetic patients, the same was observed using high glucose medium. No effect was observed in the medium with increased concentrations of insulin. CTSL knockout abolished the glucose-dependent increase in infection.

Increased glucose levels did not correlate with an increase in CTSL transcription. Rather He et al. could show that high glucose levels led to CTSL translocation from the ER into the lysosome. It was the glucose-dependent processing of the protease to its active form which promoted infection.

Overall, it is a very complete study starting from a clinical observation and ending on the molecular mechanism. A strength is certainly the wide selection of experiments. The clinical study to investigate the effect of glucose on CTSL concentrations in healthy individuals sets the stage for experiments in cell culture, animal models and human tissue. The effect of CTSL knockout cell lines on glucose-induced SARS-CoV2 infection rates are convincing. Finally, the team used a combination of Western blots and confocal microscopy to identify the underlying molecular mechanisms.

The authors keep the diabetic condition at the center of their study and extend on previous knowledge of glucose-induced CTSL activation and their consequences for Covid19 infections. By doing so, they create a novel connection between CTSL involvement in SARS-CoV2 infections and diabetes. This enables novel, public awareness of the susceptibility of diabetes patients to the disease.

---

## [Referee Report · Reviewer #2 (Public review)]

Summary:

In this study, the authors hypothesized that individuals with diabetes have elevated blood CTSL levels, which facilitates SARS-CoV-2 infection. The authors conducted in vitro experiments, revealing that elevated glucose levels promote SARS-CoV-2 infection in wild-type cells. In contrast, CTSL knockout cells show reduced susceptibility to high glucose-promoted effects. Additionally, the authors utilized lung tissue samples obtained from both diabetic and non-diabetic patients, along with db/db diabetic and control mice. Their findings indicate that diabetic conditions lead to an elevation in CTSL activity in both human and mice.

Strengths:

The authors have effectively met their research objectives, and their conclusions are supported by the data presented. Their findings suggest that high glucose levels promote CTSL maturation and translocation from the endoplasmic reticulum to the lysosome, potentially contributing to diabetic comorbidities and complications.

Weaknesses:

(1) In Figure 1e, the authors measured plasma levels of COVID-19 related proteins, including ACE2, CTSL, and CTSB, in both diabetic and non-diabetic COVID-19 patients. Notably, only CTSL levels exhibited a significant increase in diabetic patients compared to non-diabetic patients, and these levels varied throughout the course of COVID-19. Given that the diabetes groups encompass both male and female patients, it is essential to ascertain whether the authors considered the potential impact of gender on CTSL levels. The diabetes groups comprised a higher percentage of male patients (61.3%) compared to the non-diabetes group, where males constituted only 38.7%.

(2) lines145-149: "The results showed that WT Huh7 cell cultured in high glucose medium exhibited a much higher infective rate than those in low glucose medium. However, CTSL KO Huh7 cells maintained a low infective rate of SARS-CoV-2 regardless of glucose or insulin levels (Fig. 3f-h). Therefore, hyperglycemia enhanced SARS-CoV-2 infection dependent on CTSL." However, this evidence may be insufficient to support the claim that hyperglycemia enhances SARS-CoV-2 infection dependent on CTSL. The human hepatoma cell line Huh7 might not be an ideal model to validate the authors' hypothesis regarding high blood glucose promoting SARS-CoV-2 infection through CTSL.

(3) The Abstract and Introduction sections lack effective organization.

In this revised version of the study, the authors have addressed my concerns by providing additional experiments, references and discussing further the points of controversy. I think that the authors have made improvements to the manuscript.

---

## [Author Response]

The following is the authors’ response to the original reviews.

**eLife assessment**
This important study advances our understanding of why diabetes is a risk factor for more severe Covid-19 disease. The authors offer solid evidence that cathepsin L is more active in diabetic individuals, that this higher activity is recapitulated at the cellular level in the presence of high glucose, and that high glucose leads to higher cathepsin L maturation. While not all aspects of the relationship between diabetes and cathepsin L (e.g., effects of metabolic acidosis) have been investigated, the work should be of interest to researchers in diabetes, virology, and immunology.
**Public Reviews:**

**Reviewer #1 (Public Review):**
Summary:The study by He et al. investigates the relationship of an increased susceptibility of diabetes patients to COVID-19. The paper raises the possibility that hyperglycemia-induced cathepsin L maturation could be one of the driving forces in this pathology, suggesting that an increased activity of CTSL leads to accelerated virus infection rates due to an elevated processing of the SARS-CoV-2 spike protein.In a clinical case-control study, the team found that the severity of corona infections was higher in diabetic patients, and their CTSL levels correlated well with the progression of the disease. They further showed an increase in CTSL activity in the long term as well as acute hyperglycemia. SARS-CoV-2 increasingly infected cells that were cultured in serum from diabetic patients, the same was observed using high glucose medium. No effect was observed in the medium with increased concentrations of insulin. CTSL knockout abolished the glucose-dependent increase in infection.Increased glucose levels did not correlate with an increase in CTSL transcription. Rather He et al. could show that high glucose levels led to CTSL translocation from the ER into the lysosome. It was the glucose-dependent processing of the protease to its active form which promoted infection.Strengths:It is a complete study starting from a clinical observation and ending on the molecular mechanism. A strength is certainly the wide selection of experiments. The clinical study to investigate the effect of glucose on CTSL concentrations in healthy individuals sets the stage for experiments in cell culture, animal models, and human tissue. The effect of CTSL knockout cell lines on glucose-induced SARS-CoV2 infection rates is convincing. Finally, the team used a combination of Western blots and confocal microscopy to identify the underlying molecular mechanisms. The authors manage to keep the diabetic condition at the center of their study and therefore extend on previous knowledge of glucose-induced CTSL activation and their consequences for COVID-19 infections. By doing so, they create a novel connection between CTSL involvement in SARS-CoV2 infections and diabetes.Weaknesses:(1) The authors suggest that hyperglycemia as a symptom of diabetes leads to an increased infection rate in those patients. Throughout their study, the team focuses on two select symptoms of a diabetic condition, hyperglycemia and hyperinsulinemia. The team acknowledges in the discussion that there could be various other reasons. Hyperglycemia can lead to metabolic acidosis and a shift in blood pH. As CTSL activity is highly dependent on pH, it would have been crucial to include this parameter in the study.

We sincerely appreciate your valuable comment. We agree that hyperglycemia can lead to metabolic acidosis and alter blood pH. However, the normal range for blood pH in humans is relatively narrow, typically ranging from 7.35 to 7.45. In our study, we ensured that blood pH remained within this normal range for both diabetic and healthy control samples. To address your concern, we conducted experiments to investigate CTSL activity in response to pH fluctuations within this physiological range. The updated Fig. 4a now presents these findings, demonstrating consistent CTSL activity despite pH variations. Statistical analysis was performed using one-way ANOVA with Tukey’s post hoc test to ensure robustness. We have also amended the figure legend and provided corresponding descriptions in the final edition manuscript (line 15-18, page 7).

(2) The study rarely differentiates between cellular and extracellular CTSL activity. A more detailed explanation for the connection between the intracellular CTSL and serum CTSL in diabetic individuals, presumably via lysosomal exocytosis, could be helpful with regard to the final model to give a more complete picture.

Thank you for your insightful comments. Previous studies have elucidated the process by which lysosomal CTSL is transported via vesicles and subsequently secreted from the cell membrane through exocytosis (references 1-5). To provide a more comprehensive understanding, we have incorporated this information on Fig. 6h, page 32 of the final edition manuscript. This addition aims to enhance clarity regarding the connection between intracellular and serum CTSL activity in diabetic individuals, particularly through lysosomal exocytosis.

**Author response image 2. sa3fig2:** 

References：

(1) Reddy A et al. Plasma membrane repair is mediated by Ca(2+)-regulated exocytosis of lysosomes. Cell. 2001 Jul 27;106(2):157-69. doi: 10.1016/s0092-8674(01)00421-4. PMID: 11511344.

(2) Hasanagic M et al. Different Pathways to the Lysosome: Sorting out Alternatives. Int Rev Cell Mol Biol. 2015;320:75-101. doi: 10.1016/bs.ircmb.2015.07.008. Epub 2015 Aug 19. PMID: 26614872.

(3) Reiser J et al. Specialized roles for cysteine cathepsins in health and disease. J Clin Invest. 2010 Oct;120(10):3421-31. doi: 10.1172/JCI42918. Epub 2010 Oct 1. PMID: 20921628; PMCID: PMC2947230.

(4) Jaiswal JK et al. Membrane proximal lysosomes are the major vesicles responsible for calcium-dependent exocytosis in nonsecretory cells. J Cell Biol. 2002 Nov 25;159(4):625-35. doi: 10.1083/jcb.200208154. Epub 2002 Nov 18. PMID: 12438417; PMCID: PMC2173094.

(5) Coutinho MF et al. Mannose-6-phosphate pathway: a review on its role in lysosomal function and dysfunction. Mol Genet Metab. 2012 Apr;105(4):542-50. doi: 10.1016/j.ymgme.2011.12.012. Epub 2011 Dec 23. PMID: 22266136.

(3) In the early result section, an effect of hyperglycemia on total CTSL concentrations is described, but the data is not very convincing. Over the course of the manuscript, the hypothesis shifts increasingly towards an increase in protease trans-localization and processing to the active form rather than a change in total protease amounts. The overall importance of CTSL concentrations remains questionable.

Thank you for your insightful feedback. We have addressed your concerns regarding the impact of hyperglycemia on CTSL concentrations. Fig. 2h-j illustrate the effect of acute hyperglycemia on both CTSL concentration and activity in 15 healthy male volunteers over a 160-minute period. During this short timeframe, CTSL concentration remained stable, as evidenced by consistent RNA results from cells exposed to varying glucose levels (Supplementary Fig.1). However, there was a significant increase in CTSL activity, indicating that glucose elevation rapidly triggers CTSL maturation through propeptide cleavage. This activation process occurs more rapidly than CTSL protein synthesis. In summary, acute hyperglycemia specifically elevates CTSL activity, while chronic hyperglycemia may impact both CTSL activity and concentration (Fig. 2a-d). Additionally, Tournu C, et al. (1998) (reference 1) and Shi Q, et al. (2018) (reference 2) have reported that increased glucose metabolism promotes the maturation and secretion of CTSL and other proteases. These findings align with our evidence that hyperglycemia drives CTSL maturation, as discussed at line 10-25, page 12 in the final edition manuscript.

References：

(1) Tournu C et al. Glucose controls cathepsin expression in Ras-transformed fibroblasts. Arch Biochem Biophys. 1998 Dec 1;360(1):15-24. doi: 10.1006/abbi.1998.0916. PMID: 9826424.

(2) Shi Q et al. Increased glucose metabolism in TAMs fuels O-GlcNAcylation of lysosomal Cathepsin B to promote cancer metastasis and chemoresistance. Cancer Cell. 2022 Oct 10;40(10):1207-1222.e10. doi: 10.1016/j.ccell.2022.08.012. Epub 2022 Sep 8. PMID: 36084651.

**Reviewer #2 (Public Review):**
Summary:In this study, the authors hypothesized that individuals with diabetes have elevated blood CTSL levels, which facilitates SARS-CoV-2 infection. The authors conducted in vitro experiments, revealing that elevated glucose levels promote SARS-CoV-2 infection in wild-type cells. In contrast, CTSL knockout cells show reduced susceptibility to high glucose-promoted effects. Additionally, the authors utilized lung tissue samples obtained from both diabetic and non-diabetic patients, along with db/db diabetic and control mice. Their findings indicate that diabetic conditions lead to an elevation in CTSL activity in both humans and mice.Strengths:The authors have effectively met their research objectives, and their conclusions are supported by the data presented. Their findings suggest that high glucose levels promote CTSL maturation and translocation from the endoplasmic reticulum to the lysosome, potentially contributing to diabetic comorbidities and complications.Weaknesses:(1) In Figure 1e, the authors measured plasma levels of COVID-19 related proteins, including ACE2, CTSL, and CTSB, in both diabetic and non-diabetic COVID-19 patients. Notably, only CTSL levels exhibited a significant increase in diabetic patients compared to non-diabetic patients, and these levels varied throughout the course of COVID-19. Given that the diabetes groups encompass both male and female patients, it is essential to ascertain whether the authors considered the potential impact of gender on CTSL levels. The diabetes groups comprised a higher percentage of male patients (61.3%) compared to the non-diabetes group, where males constituted only 38.7%.

Thank you for your insightful feedback. In response to your concerns regarding the potential impact of gender on CTSL levels in diabetic and non-diabetic COVID-19 patients, we conducted analyses to address this issue. While our initial study involved 62 COVID-19 patients, with 31 having diabetes and 31 without, matching based on gender and age, we acknowledged the challenge of obtaining balanced gender distribution in both groups due to the difficulty of collecting blood samples from COVID-19 patients. To mitigate potential gender bias resulting from small sample sizes, we conducted a supplementary clinical study involving 122 non-COVID-19 volunteers, including 61 individuals with diabetes and 61 without. The percentage of males in the diabetes group was 50.8%, while in the healthy group, males constituted 44.3% (P value = 0.468), indicating no significant gender bias. We have incorporated this information into the discussion section on line 4-13, page 11 in the final edition manuscript, to provide clarity on this aspect of our study.

(2) Lines 145-149: "The results showed that WT Huh7 cell cultured in high glucose medium exhibited a much higher infective rate than those in low glucose medium. However, CTSL KO Huh7 cells maintained a low infective rate of SARS-CoV-2 regardless of glucose or insulin levels (Fig. 3f-h). Therefore, hyperglycemia enhanced SARS-CoV-2 infection dependent on CTSL." However, this evidence may be insufficient to support the claim that hyperglycemia enhances SARS-CoV-2 infection dependent on CTSL. The human hepatoma cell line Huh7 might not be an ideal model to validate the authors' hypothesis regarding high blood glucose promoting SARS-CoV-2 infection through CTSL.

Thank you for your valuable feedback. We have addressed the concerns regarding the sufficiency of evidence supporting the claim that hyperglycemia enhances SARS-CoV-2 infection dependent on CTSL. Specifically, we have revised the expression to state, “Therefore, hyperglycemia enhanced SARS-CoV-2 infection through CTSL.” as suggested, in line 9, page 7 in the final edition manuscript. Additionally, we acknowledge the potential involvement of other bioactive factors, such as 1,5-anhydro-D-glucitol (1,5-AG), in mediating SARS-CoV-2 infection in patients with diabetes, as outlined in the discussion section from line 13-21, page 13 in the final edition manuscript.

Regarding the choice of the human hepatoma cell line Huh7 as a model for investigating hyperglycemia-induced CTSL maturation and SARS-CoV-2 infection, we recognize the importance of tissue specificity and the liver’s significance as a target organ for COVID-19. Despite potential limitations, such as generalization of liver function abnormalities and lack of tissue specificity in SARS-CoV-2 impact, Huh7 cells offer practical advantages as a mature cell model for studying SARS-CoV-2 infection, including accessibility, susceptibility to infection, and stable proliferation (reference 1-3). We have elaborated on these considerations in the discussion section at line 19-23, page 11 in the final edition manuscript, to provide context for our choice of experimental model.

References：

(1) Gupta A et al. Extrapulmonary manifestations of COVID-19. Nat Med. 2020 Jul;26(7):1017-1032. doi: 10.1038/s41591-020-0968-3. Epub 2020 Jul 10. PMID: 32651579.

(2) Nie X et al. Multi-organ proteomic landscape of COVID-19 autopsies. Cell. 2021 Feb 4;184(3):775-791.e14. doi: 10.1016/j.cell.2021.01.004. Epub 2021 Jan 9. PMID: 33503446; PMCID: PMC7794601.

(3) Ciotti M et al. The COVID-19 pandemic. Crit Rev Clin Lab Sci. 2020 Sep;57(6):365-388. doi: 10.1080/10408363.2020.1783198. Epub 2020 Jul 9. PMID: 32645276.

(3) The Abstract and Introduction sections lack effective organization.

Thank you for your valuable comments. We have rewritten the Abstract and Introduction sections and incorporated the updated descriptions in the final edition manuscript.

**Reviewer #1 (Recommendations For The Authors):**
(1) When referring to diabetes, does this exclusively include diabetes type 2?

Thank you for your inquiry. In our study, the term “diabetes” encompasses the condition of hyperglycemia in a broad sense, rather than specifically indicating type 1 diabetes (T1DM) or type 2 diabetes (T2DM). This broader definition aligns with the scope of our research objectives and findings, particularly observed in the cell experiments conducted. We have clarified this point in the revised discussion section, from line 6-9, page 12 in the final edition manuscript, to provide additional context for readers.

(2) The titles of the individual paragraphs are not very strong and descriptive. More precise titles help to structure the paper better for the reader.

Thank you for your valuable comments. We have rewritten the title of each section to make it more precise for readers and incorporated the updated descriptions in the manuscript.

(3) Fig.3c, adding a 0 nM insulin control would be nice.

Thank you for your suggestion. We have revised Fig.3c according to your advice. The revised figure was located at page 29 in the final edition manuscript. The corresponding figure legend has also been revised.

**Author response image 3. sa3fig3:** 

(4) Fig.3e non-infection control would be nice.

Thank you for your suggestion. We have incorporated your feedback by adding a non-infection control in Fig. 3e. In this revised figure, we included a measurement of SARS-CoV-2 pseudovirus infection assessed through the fluorescence captured by a reader. Cells infected by the pseudovirus exhibited activation of the firefly luciferase, resulting in the release of fluorescence. Conversely, non-infected control cells showed no fluorescence, with the reader recording a value of zero. The updated figure can now be found on page 29 in the final edition manuscript, and we have adjusted the corresponding figure legend accordingly.

**Author response image 4. sa3fig4:** 

(5) In Figure 5, the processing of CTSL in cells (b-c) strongly differs from processing in tissue (d-e) focusing on amounts of dc-mCTSL. Do you have an explanation for this? Overall, blots are hard to judge by eye and it would be nice to include blots with shorter exposure.

Thank you for your insightful feedback. The differences observed in the processing of CTSL between cells (Fig. 5b) and tissues (Fig. 5d-e) may be attributed to the complexities inherent in tissue samples, which can impact the clarity of the images. Furthermore, in human tissue samples, it is pertinent to consider that patients in the diabetes group had their blood glucose levels controlled within or near the normal range prior to lung surgery. As a result, the evidence supporting CTSL maturation in human lung tissue blotting images may be less compelling. We have addressed this aspect in the revised results section (lines 10-13, page 9). Additionally, we will consider including blots with shorter exposure to enhance visual clarity in future studies.

(6) Considering Fig2B and Figure S1, the evidence of an effect of hyperglycemia or high glucose medium on total CTSL protein concentration is not very strong. In my opinion, this claim in the results section for Fig2 should be revisited.

Thank you for your valuable suggestion. We have revisited the section in question and made appropriate revisions. The original sentence has been modified to accurately reflect the findings: "We found that plasma CTSL activity was strongly positively correlated with chronic hyperglycemia indicated by HbA1c and was significantly higher in diabetic patients than in euglycemic individuals (Fig. 2a, c). Additionally, plasma CTSL concentration showed a positive trend with chronic hyperglycemia indicated by HbA1c (Fig. 2b, d)". These changes have been incorporated into the revised results section (lines 12-16, page 5).

(7) Overall, data hinting to increased CTSL activity is stronger than protein amount. This being said, in hyperglycemia, blood pH can be affected (metabolic acidosis). As CTSL has higher activity at low pH, could the increase in activity be caused by a drop in pH? Can you include this aspect in your manuscript? For example, is there a pH difference in serum of nondiabetic vs diabetic patients?

Thank you for your valuable input. We have already addressed the potential impact of pH changes on CTSL activity in our response to Weakness No. 1. As indicated, although hyperglycemia can lead to metabolic acidosis and changes in blood pH, the pH levels observed in our study remained within the normal range (7.35 to 7.45). Therefore, we conducted experiments to investigate CTSL activity in response to changes in pH, which showed consistent activity levels within this range. This information has been included in our revised manuscript (line 15-18, page 7).

**Reviewer #2 (Recommendations For The Authors):**
(1) The Abstract and Introduction sections lack effective organization. The manuscript's style resembles that of Cell Journal rather than aligning with the customary format of eLife.

Thank you for your valuable comments. The Abstract and Introduction sections have been reorganized to be more precise for readers has been included in our revised manuscript. Additionally, we have meticulously updated the manuscript's style to align with the standard format of eLife in our revised manuscript, especially key resources table of materials and methods sections.